# Quantum Hairy Black Hole Formation and Horizon Quantum Mechanics

**Rogerio Teixeira Cavalcanti** *,† and **Julio Marny Hoff da Silva** †

Departamento de Física, Universidade Estadual Paulista (Unesp), Guaratinguetá 12516-410, Brazil
* Correspondence: rogerio.cavalcanti@unesp.br
† These authors contributed equally to this work.

**Abstract:** After introducing the gravitational decoupling method and the hairy black hole recently derived from it, we investigate the formation of quantum hairy black holes by applying the horizon quantum mechanics formalism. It enables us to determine how external fields, characterized by hairy parameters, affect the probability of spherically symmetric black hole formation and the generalized uncertainty principle.

**Keywords:** gravitational decoupling; hairy black holes; quantum black holes; horizon quantum mechanics; generalized uncertainty principle

## 1. Introduction

Given their intrinsic connection with intense gravitational fields, solid theoretical basis [1–3], and several observational results corroborating their existences, black holes play a central role in contemporary high-energy physics and astrophysics [4–7]. Despite the characterization of the horizon of stationary black hole solutions being well-known within general relativity [3,8], the nature of the horizons of non-stationary or stationary solutions beyond general relativity is still a source of extensive research [9–12]. The investigation of black holes is not restricted to astrophysical objects; they are also expected to be formed whenever a high concentration of energy is confined to a small region of spacetime, producing so-called quantum black holes [7,13–17]. However, the precise formation mechanism of classical and quantum black holes is still unknown. Although we do not have a theory of quantum gravity, phenomenology suggests that some features of quantum black holes are expected to be model-independent [7]. From a certain scale, candidate theories should modify the results of general relativity, giving birth to some alternatives to Einsteins's theory of gravity [18,19]. Examples could allow for the presence of non-minimal coupled fundamental fields or higher derivative terms during the action, which directly affects the uniqueness theorems of black holes in general relativity. The famous no-hair theorem is not preserved outside the general relativity realm. These solutions lead to effects that are potentially detectable near the horizon of astrophysical black holes [20–22], or in quantum black holes' formation [23,24], and may provide hints for the quantum path.

One of the major challenges in general relativity is finding physically relevant solutions to Einstein's field equations. On the other hand, deriving new solutions from other previously known ones is a widespread technique. This approach is precisely what the so-called gravitational decoupling (GD) method intends to achieve. It has recently commanded the community's attention due to its simplicity and effectiveness [25–27] in generating new, exact analytical solutions by considering additional sources to the stress-energy tensor. The recent description of anisotropic stellar distributions [28,29], whose predictions might be tested in astrophysical observations [30–33], as well as the hairy black hole solutions by gravitational decoupling, are particularly interesting. The latter describes a black hole with hair sourced by generic fields, possibly of quantum nature, surrounding the vacuum

Schwarzschild solution [27]. Exciting results have been found during investigation of this solution [34–36].

From the quantum side, one of the key features of quantum gravity phenomenology is the generalized uncertainty principle (GUP), which modifies the Heisenberg uncertainty principle accordingly

$$\Delta x \Delta p \gtrsim \hbar \left(1 + \epsilon (\Delta p)^2\right). \tag{1}$$

This expression of the GUP, which stems from different approaches to quantum gravity [37–46], characterizes a minimum scale length $\Delta x$. This feature emerges quite naturally in the horizon quantum mechanics formalism (HQM) [16,47]. In addition to the GUP, HQM also provides an estimation of the probability of quantum black hole formation. In a scenario of extra-dimensional spacetimes, the HQM gave an explanation for the null results of quantum black hole formation in current colliders [23,24]. Could it also tell us something about a mechanism for decreasing the fundamental scale to something near the scale of current colliders? Our aim is to investigate the quantitative and qualitative effects of black hole hair, regarding the probability of black hole formation and the GUP by applying the horizon quantum mechanics formalism.

This paper is organized as follows: Section 2 is dedicated to reviewing the gravitational decoupling procedure, the metric for GD hairy black holes, and an approximation for the horizon radius. In Section 3, we apply the horizon quantum mechanics formalism to the hairy black hole solution of the previous section. We compare the probability of quantum black hole formation and the GUPs of hairy black holes for a range of hair parameters, unveiling the effects of the hair fields. Finally, Section 4 is dedicated to conclusions and discussion.

## 2. Hairy Black Holes and Horizon Radius

Starting from Einstein's field equations

$$G_{\mu\nu} = 8\pi \check{T}_{\mu\nu}, \tag{2}$$

where $G_{\mu\nu} = R_{\mu\nu} - \frac{1}{2}Rg_{\mu\nu}$ denotes the Einstein tensor, the gravitational decoupling (GD) [25] method takes the energy–momentum tensor decomposed as

$$\check{T}_{\mu\nu} = T_{\mu\nu} + \Theta_{\mu\nu}. \tag{3}$$

Here, $T_{\mu\nu}$ is the source of a known solution to general relativity, while $\Theta_{\mu\nu}$ introduces a new field or extension of the gravitational sector. From $\nabla_\mu G^{\mu\nu} = 0$, we also have $\nabla_\mu \check{T}^{\mu\nu} = 0$. The effective density and the tangential and radial pressures can be determined by examining the field equations

$$\check{\rho} = \rho + \Theta_0^0, \tag{4a}$$
$$\check{p}_t = p - \Theta_2^2, \tag{4b}$$
$$\check{p}_r = p - \Theta_1^1. \tag{4c}$$

The idea is to deform a known solution to split the field equations in a sector containing the known solution with source $T_{\mu\nu}$ and a decoupled one governing the deformation, encompassing $\Theta_{\mu\nu}$. In fact, assuming a known spherically symmetric metric,

$$ds^2 = -e^{\kappa(r)}dt^2 + e^{\zeta(r)}dr^2 + r^2 d\Omega^2, \tag{5}$$

and deforming $\kappa(r)$ and $\zeta(r)$ as

$$\kappa(r) \mapsto \kappa(r) + \alpha f_2(r), \tag{6a}$$
$$e^{-\zeta(r)} \mapsto e^{-\zeta(r)} + \alpha f_1(r), \tag{6b}$$

the resulting decoupled field equations read

$$8\pi\,\Theta_0^{\,0} \;=\; \alpha\left(\frac{f_1}{r^2} + \frac{f_1'}{r}\right), \tag{7a}$$

$$8\pi\,\Theta_1^{\,1} - \alpha\,\frac{e^{-\zeta}\,f_2'}{r} \;=\; \alpha f_1\left(\frac{1}{r^2} + \frac{\kappa'(r) + \alpha f_2'(r)}{r}\right), \tag{7b}$$

$$8\pi\Theta_2^{\,2} - \alpha f_1 Z_1(r) \;=\; \alpha\frac{f_1'}{4}\left(\kappa'(r) + \alpha f_2'(r) + \frac{2}{r}\right) + \alpha Z_2(r), \tag{7c}$$

where [25]

$$Z_1(r) \;=\; \alpha^2 f_2'(r)^2 + 2\,\alpha\left(f_2'(r)\kappa'(r) + \frac{f_2'(r)}{r} + f_2''(r)\right) + \kappa'(r)^2 + \frac{2\,\kappa'(r)}{r} + 2\,\kappa''(r), \tag{8a}$$

$$Z_2(r) \;=\; \alpha e^{-\zeta}\left(2f_2'' + f_2^{2\prime} + \frac{2f_2'}{r} + 2\kappa' f_2' - \zeta' f_2'\right). \tag{8b}$$

The above equations state that if the deformation parameter $\alpha$ goes to zero, then $\Theta_{\mu\nu}$ must go to zero. It is worth mentioning that for extended geometric deformation, that is, for $f_2 \neq 0$, the sources are not individually conserved in general. However, as discussed in [26], in this case, the decoupling of the field equations without an exchange of energy is allowed in two scenarios: (a) when $T_{\mu\nu}$ is a barotropic fluid whose equation of state is $T_0^{\,0} = T_1^{\,1}$ or (b) for vacuum regions of the first system $T_{\mu\nu} = 0$. When minimal geometric deformation is applied, on the other hand, the sources are shown to be individually conserved [25,26].

Assuming the Schwarzschild solution to be the known one and requiring a well-defined horizon structure [27], from $g_{rr} = -\frac{1}{g_{tt}}$ follows

$$\left(1 - \frac{2M}{r}\right)\left(e^{\alpha f_2(r)} - 1\right) = \alpha f_1(r). \tag{9}$$

Therefore, one is able to write

$$ds^2 \;=\; -\left(1 - \frac{2M}{r}\right)e^{\alpha f_2(r)}dt^2 + \left(1 - \frac{2M}{r}\right)^{-1}e^{-\alpha f_2(r)}dr^2 + r^2\,d\Omega^2. \tag{10}$$

Further, assuming strong energy conditions,

$$\check{\rho} + \check{p}_r + 2\,\check{p}_t \geq 0, \tag{11a}$$
$$\check{\rho} + \check{p}_r \geq 0, \tag{11b}$$
$$\check{\rho} + \check{p}_t \geq 0, \tag{11c}$$

and managing the field equations, a new hairy black hole solution was found [27]

$$ds^2 = -f(r)dt^2 + \frac{1}{f(r)}dr^2 + r^2 d\Omega^2, \tag{12}$$

where

$$f(r) = 1 - \frac{2GM + \alpha\ell}{r} + \alpha e^{-\frac{r}{GM}}. \tag{13}$$

The dimensionless parameter $0 \leq \alpha \leq 1$ tracks the deformation of the Schwarzschild black hole, $e$ is the Euler constant, and $\ell$ is the direct effect of the nonvanishing additional font $\Theta_{\mu\nu}$. Notice that by taking $\alpha = 0$, the Schwarzschild solution is restored. Further, the $\ell$ parameter is limited to $2GM/e^2 \leq \ell \leq 1$ due to the assumption of a strong energy condition. In extreme cases, $\ell = 2GM/e^2$ and

$$f_e(r) = 1 - \frac{2GM}{r} + \alpha\left(e^{-\frac{r}{GM}} - \frac{2GM}{e^2\,r}\right). \tag{14}$$

The hairy black hole has a single horizon, located at $r = r_H$, such that

$$\left(1 + \alpha e^{-\frac{r_H}{GM}}\right) r_H = 2GM + \alpha\ell. \tag{15}$$

Such an equation has no analytical solution. Nevertheless, a very accurate analytical approximation is found by Taylor expanding it around the Schwarzschild horizon radius $r_S = 2GM$,

$$\frac{r_H}{GM} \approx \frac{4\left(\alpha\ell e^2/GM - 3\alpha + e^2\right)}{\alpha\ell e^2/GM - 4\alpha + 2e^2}. \tag{16}$$

Figure 1 shows a comparison between the exact and approximated horizon radii for different values of the hairy parameters. In the following section, we are going to use Equation (16) for the analytical expression of the hairy black hole's horizon radius.

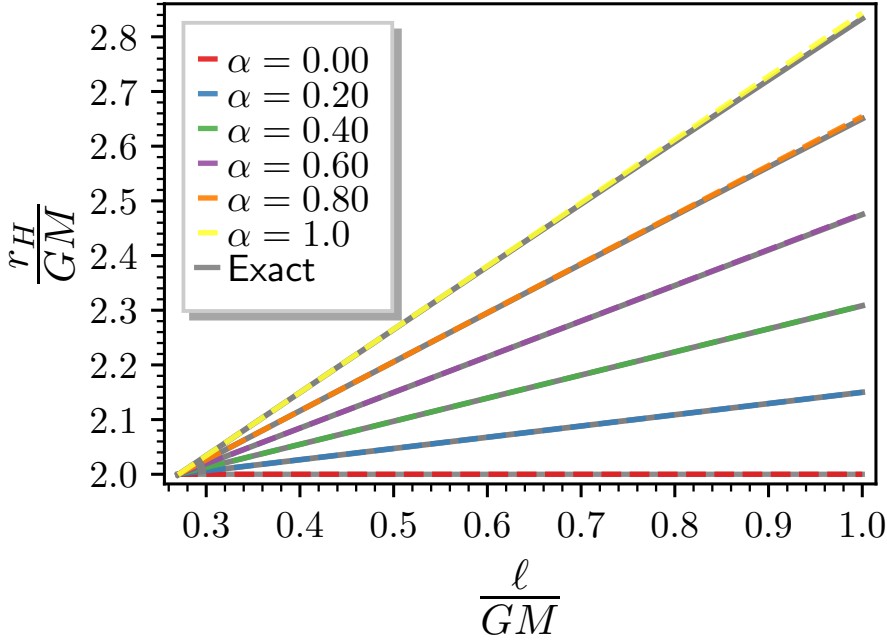

**Figure 1.** The radius of the hairy black hole horizon $r_H$ as a function of $\ell$ for different values of the parameter $\alpha$. The colored dashed lines represent the approximated radius, and the gray lines are the exact ones. It shows how the hairy horizon deviates from the Schwarzschild horizon for an increasing $\alpha$ and $\ell$. The ranges for $\alpha$ and $\ell$ were fixed due to the assumption of a strong energy condition [27].

### 3. The Horizon Quantum Mechanics Formalism

Horizon quantum mechanics (also known as horizon wave function formalism) is an effective approach capable of providing the signatures of black hole physics to the Planck scale [48–51] (see [47] for a comprehensive review). The main idea is to extend quantum mechanics and gravity further than the current experimental limits. In such an approach, we face the conceptual challenge of consistently describing classical and quantum mechanical objects, such as horizons and particles. This is achieved by assigning wave functions to the quantum black hole horizon. This association allows the use of quantum mechanical machinery to distinguish between particles and quantum black holes and to estimate the GUPs. Nevertheless, first, we must choose a model describing the particle wave function to derive the results. Due to the previous results' simplicity and efficiency, we shall use the Gaussian model.

From classical general relativity, we know that the horizons of black holes are described by trapping surfaces, whose locations are determined by

$$g^{ij}\nabla_i r \nabla_j r = 0, \tag{17}$$

where $\nabla_i r$ is orthogonal to the surfaces of the constant area $\mathcal{A} = 4\pi r^2$. A trapping surface then exists if there are values of $r$ and $t$ such that the gravitational radius $R_H$ satisfies

$$R_H(r,t) \geq r \,. \tag{18}$$

Considering a spinless point-particle of mass $m$, an uncertainty in the spatial particle localization of the same order of the Compton scale $\lambda_m \simeq \hbar/m = l_p \, m_p/m$ follows from the uncertainty principle, where $l_p$ and $m_p$ are the Planck length and mass, respectively. Arguing that quantum mechanics gives a more precise description of physics, $R_H$ makes sense only if it is larger than the Compton wavelength associated with the same mass, namely $R_H \gtrsim \lambda_m$. Thus, for the Schwarzschild radius $R_S = 2Gm = 2\frac{l_p}{m_p}m$,

$$l_p \, m/m_p \gtrsim l_p \, m_p/m \quad \Longrightarrow \quad m \gtrsim m_p \,. \tag{19}$$

This suggests that the Planck mass is the minimum mass such that the Schwarzschild radius can be defined.

From quantum mechanics, the spectral decomposition of a spherically symmetric matter distribution is given by the expression

$$|\psi_S\rangle = \sum_E C(E)|\psi_E\rangle \,, \tag{20}$$

with the usual eigenfunction equation

$$\hat{H}|\psi_E\rangle = E|\psi_E\rangle \,, \tag{21}$$

regardless of the specific form of the actual Hamiltonian operator $\hat{H}$. Using the energy spectrum and inverting the expression of the Schwarzschild radius, we have

$$E = m_p \frac{r_H}{2l_p}. \tag{22}$$

Putting it back into the wave function, one can define the (unnormalized) horizon wave function as

$$\psi_H(r_H) = C\left(m_p \frac{r_H}{2l_p}\right) \tag{23}$$

whose normalization is fixed, as usual, by the inner product

$$\langle\psi_H|\phi_H\rangle = 4\pi \int_0^\infty \psi_H^*(r_H)\phi_H(r_H)r_H^2 dr_H. \tag{24}$$

However, the classical radius $R_H$ is thus replaced by the expected value of the operator $\hat{R}_H$. From the uncertainty of the expectation value, it follows that the radius will necessarily be "fuzzy", similar to the position of the source itself. The next aspect one has to approach to establish a criterion for deciding if a mass distribution does or does not form a black hole is if it lies inside its horizon of radius $r = r_H$. From quantum mechanics, one finds that it is given by the product

$$\mathcal{P}_<(r < r_H) = P_S(r < r_H)\mathcal{P}_H(r_H), \tag{25}$$

where the first term,

$$P_S(r < r_H) = 4\pi \int_0^{r_H} |\psi_S(r)|^2 r^2 dr, \tag{26}$$

is the probability that the particle resides inside the sphere of radius $r = r_{\mathrm{H}}$, while the second term,

$$\mathcal{P}_H(r_{\mathrm{H}}) = 4\pi r_{\mathrm{H}}^2 |\psi_H(r_{\mathrm{H}})|^2 \tag{27}$$

is the probability density that the value of the gravitational radius is $r_{\mathrm{H}}$. Finally, the probability that the particle described by the wave function $\psi_S$ is a BH will be given by the integral of (25) over all possible values of the horizon radius $r_{\mathrm{H}}$. Namely,

$$P_{BH} = \int_0^\infty \mathcal{P}_<(r < r_{\mathrm{H}}) dr_{\mathrm{H}}, \tag{28}$$

which is one of the main outcomes of the formalism.

*3.1. Gaussian Sources*

The previous construction can be made explicit by applying the Gaussian model for the wave function. To implement this idea, let us recall that spectral decomposition is also assumed to be valid for momentum. Therefore, from (20), $\langle p|\psi_S\rangle = C(p) \equiv \psi_H(p)$. The Gaussian wave function for $\psi_S$ scales as $r^2$ in the position space and leads to a Gaussian wave function in the momentum space, scaling as $p^2$, naturally. Finally, since the dispersion relation relates $p^2$ with energy, we are able to have $\langle p|\psi_S\rangle = \psi_H(r_H)$ via (22). Hence, starting with a Gaussian wave function, we can describe a spherically symmetric massive particle at rest, such as

$$\psi_S(r) = \frac{e^{-\frac{r^2}{2l^2}}}{(l\sqrt{\pi})^{3/2}}. \tag{29}$$

The corresponding function in momentum space is thus given by

$$\tilde{\psi}_S(p) = 4\pi \int_0^\infty \frac{\sin(rp)}{\sqrt{8\pi^3}rp} \frac{e^{-\frac{r^2}{2l^2}}}{(l\sqrt{\pi})^{3/2}} r^2 dr$$

$$= \frac{e^{-\frac{p^2}{2\Delta^2}}}{(\Delta\sqrt{\pi})^{3/2}}, \tag{30}$$

where $\Delta = m_p l_p / l$ is the spread of the wave packet in momentum space, whose width $l$ the Compton length of the particle should diminish,

$$l \geq \lambda_m \sim \frac{m_p l_p}{m}. \tag{31}$$

In addition to the straightforward handling of a Gaussian wave packet, it is also relevant to recall that the Gaussian wave function leads to a minimal uncertainty for the expected values computed with it. Had we used another wave function, it would certainly imply a worsening uncertainty, eventually leading to unnecessary extra difficulties relating to the HQM and GUP (see next section). Back to our problem, assuming the relativistic mass-shell relation in flat space [48]

$$p^2 = E^2 - m^2, \tag{32}$$

the energy $E$ of the particle is expressed in terms of the related horizon radius $r_{\mathrm{H}} = R_{\mathrm{H}}(E)$, following from Equation (16),

$$E = \frac{\alpha m_p \ell e^2 + (\alpha - e^2) m_p r_H}{2(2\alpha - e^2) l_p}. \tag{33}$$

Thus, from Equations (30) and (33), one finds the the horizon wave function of the hairy black hole

$$\psi_H(r_H) = \mathcal{N}_H \Theta(r_H - R_H)\, e^{\left(C_2 r_H^2 + C_1 r_H + C_0\right)},$$

where

$$C_0 = -\frac{\alpha^2 l^2 m_p^2 \ell^2 e^4}{8\left(2\,\alpha - e^2\right)^2 l_p^2}, \quad C_1 = -\frac{\left(\alpha - e^2\right)\alpha l^2 m_p^2 \ell e^2}{4\left(2\,\alpha - e^2\right)^2 l_p^2}, \quad C_2 = -\frac{\left(\alpha - e^2\right)^2 l^2 m_p^2}{8\left(2\,\alpha - e^2\right)^2 l_p^2}. \tag{34}$$

The Heaviside step function $\Theta$ appears above due to the imposition $E \geq m$. The normalisation factor $\mathcal{N}_H$ is fixed according to

$$\mathcal{N}_H^{-2} = 4\pi \int_0^\infty |\psi_H(r_H)|^2\, r_H^2\, dr_H.$$

The normalized horizon wave function is thus given as follows

$$\psi_H(r_H) = -\frac{2\, C_2^{\frac{3}{2}} e^{\frac{A(r_H)}{2}}}{\sqrt{\pi}\sqrt{4\, C_1 C_2 e^{A(R_H)} - \left(2\sqrt{2} C_2 \Gamma\left(\frac{3}{2}, -A(R_H)\right) + \sqrt{2\pi} C_1^2 \left(\mathrm{erf}\left(\frac{\sqrt{2}(2\, C_2 R_H + C_1)}{2\sqrt{-C_2}}\right) - 1\right)\right)\sqrt{-C_2}}}, \tag{35}$$

$$A(x) = \frac{4\, C_2^2 x^2 + 4\, C_1 C_2 x + C_1^2}{2\, C_2}.$$

Here, $\Gamma(s, x)$ denotes the upper incomplete Euler–Gamma function and $\mathrm{erf}(x)$ the error function. The expression above has two classes of parameters. Two of these, $\alpha$ and $\ell$, are related to the hairy black hole, and two are non-fixed *a priori*: the particle mass $m$, encoded in $R_H$, and the Gaussian width $l$. The resulting probability $P_{BH} = P_{BH}(l, m, \ell, \alpha)$ will also depend on the same parameters.

According to the previous discussion, before finding the probability distribution, we have first to find the probability that the particle resides inside a sphere with the radius $r = r_H$. From Equations (26) and (29), one obtains

$$P_S(r < r_H) = 4\pi \int_0^{r_H} |\psi_S(r)|^2 r^2 dr = \frac{2}{\sqrt{\pi}} \gamma\left(\frac{3}{2}, \frac{r_H^2}{l^2}\right),$$

with $\gamma(s, x) = \Gamma(s) - \Gamma(s, x)$, the lower incomplete Gamma function. Equations (27) and (35) yield $\mathcal{P}_H(r_H)$, as depicted in Figure 2.

Combining the previous results, one finds that the probability density for the particle resides within its own gravitational radius

$$\mathcal{P}_<(r < r_H) = 8\sqrt{\pi} \gamma\left(\frac{3}{2}, \frac{r_H^2}{l^2}\right) r_H^2 |\psi_H(r_H)|^2.$$

The probability of the particle described by the Gaussian to be a black hole is finally given by

$$P_{BH}(l, m, \ell, \alpha) = 8\sqrt{\pi} \int_{R_H}^\infty \gamma\left(\frac{3}{2}, \frac{r_H^2}{l^2}\right) r_H^2 |\psi_H(r_H)|^2, \tag{36}$$

which has to be calculated numerically. Assuming the Gaussian width has the same order as the particle Compton length, we could set $l \sim m^{-1}$ on Equation (36) and find the probability depending on either $l$ or $m$. On the other hand, by departing again from Equation (31), we

may set values for $m$ in terms of the Planck mass and find the probability in this scenario. Applying $l \sim m^{-1}$ yields

$$P_{BH}(l, \ell, \alpha) = 8\sqrt{\pi} \int_{R_H}^{\infty} \gamma\left(\frac{3}{2}, \frac{r_H^2}{l^2}\right) r_H^2 |\psi_H(r_H)|^2, \tag{37}$$

or

$$P_{BH}(m, \ell, \alpha) = 8\sqrt{\pi} \int_{R_H}^{\infty} \gamma\left(\frac{3}{2}, r_H^2 m^2\right) r_H^2 |\psi_H(r_H)|^2. \tag{38}$$

The resulting probabilities are shown in Figure 3 below. Figure 4 displays the probability for $m$ given as a fraction of the Planck mass.

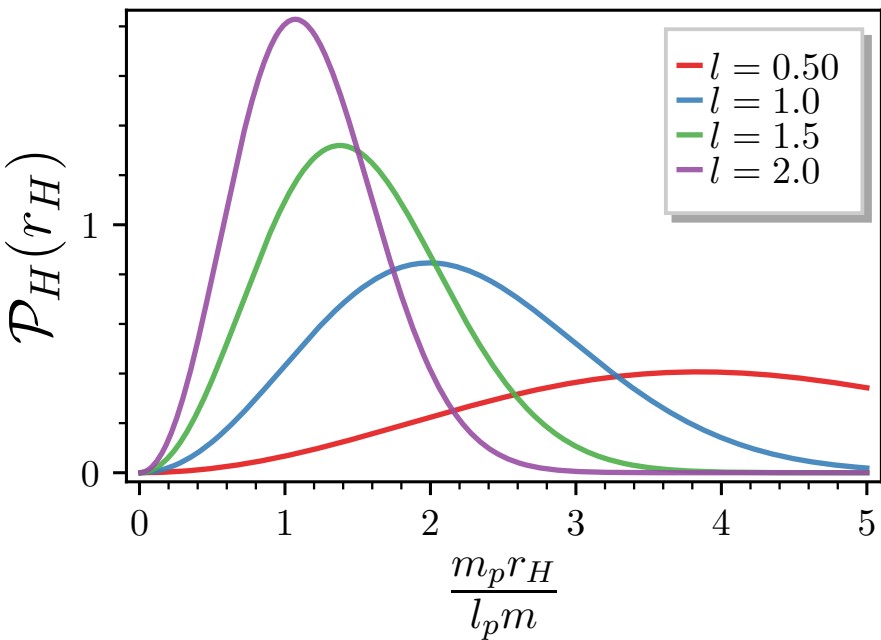

**Figure 2.** The probability density for the value of the gravitational radius is $r_H$ for $\alpha = \ell/(GM) = 0.5$ and different values of the Gaussian width.

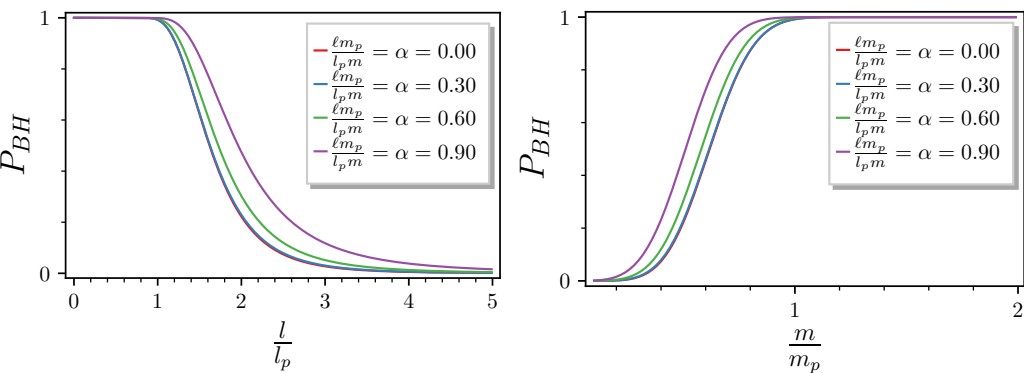

**Figure 3.** The probability of a "particle" being a black hole depending on the Gaussian width or mass, assuming $l \sim m^{-1}$.

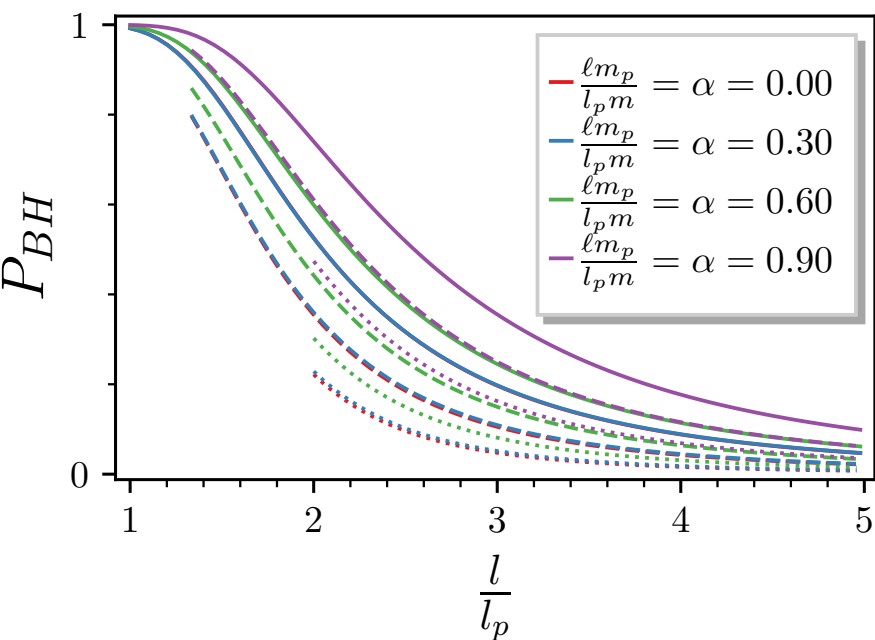

**Figure 4.** The probability of a "particle" being a black hole depending on the Gaussian width and mass $m$ given as a fraction of the Planck mass, with $m = m_p$ (solid), $m = 3m_p/4$ (dashed), and $m = m_p/2$ (dotted).

*3.2. HQM and GUP*

Since the horizon quantum mechanics formalism applies the standard wave function description for particles, a natural question is whether it affects the Heisenberg uncertainty principle. As mentioned, it produces a GUP similar to that produced by Equation (1). In quantum mechanics, the uncertainty principle may be derived by calculating the uncertainty associated with the wave function. Here, we start from the same point. From the Gaussian wave function (29), the particle size uncertainty is given by

$$
\begin{aligned}
\Delta r_0^2 &= \langle r^2 \rangle - \langle r \rangle^2 \\
&= 4\pi \int_0^\infty |\psi_S(r)|^2 r^4 dr - \left( 4\pi \int_0^\infty |\psi_S(r)|^2 r^3 dr \right)^2 \\
&= \frac{3\pi - 8}{2\pi} l^2.
\end{aligned}
\tag{39}
$$

One might find the uncertainty of the horizon radius in an analogous way,[1]

$$
\Delta r_{\mathrm{H}}^2 = \langle r_{\mathrm{H}}^2 \rangle - \langle r_{\mathrm{H}} \rangle^2.
\tag{40}
$$

The total radial uncertainty can now be taken as a linear combination of the quantities calculated above, $\Delta r = \Delta r_0 + \epsilon \Delta r_{\mathrm{H}}$. For the uncertainty in momentum, we have

$$
\Delta p^2 = \langle p^2 \rangle - \langle p \rangle^2 = \frac{3\pi - 8}{2\pi} \frac{m_p^2 l_p^2}{l^2}.
$$

Note that the momentum uncertainty and the width $l$ are related such that $\Delta p \sim 1/l$. Using this fact in $\Delta r = \Delta r_0 + \epsilon \Delta r_{\mathrm{H}}$, one is able to find

$$
\frac{\Delta r}{l_p} = \frac{3\pi - 8}{2\pi} \frac{m_p}{\Delta p} + \epsilon \Delta_{\mathrm{H}} \left( \frac{\Delta p}{m_p} \right),
\tag{41}
$$

which is similar to the GUP discussed previously. The function $\Delta_{\mathrm{H}}$ also depends on the wave function and hairy black hole parameters. Figure 5 shows the behavior of the GUP

as a function of the momentum uncertainty, taking $\epsilon = 1$. There, we can see a minimum $\Delta r$ placed around the Planck scale. From the GUP expression, it is straightforward to see that a larger $\epsilon$ means significant correction to the quantum mechanics' uncertainty. The hairy parameters, however, have a small qualitative effect on fixing the minimum scale. As shown in Figure 5, their effects become prominent for a large $\Delta p$.

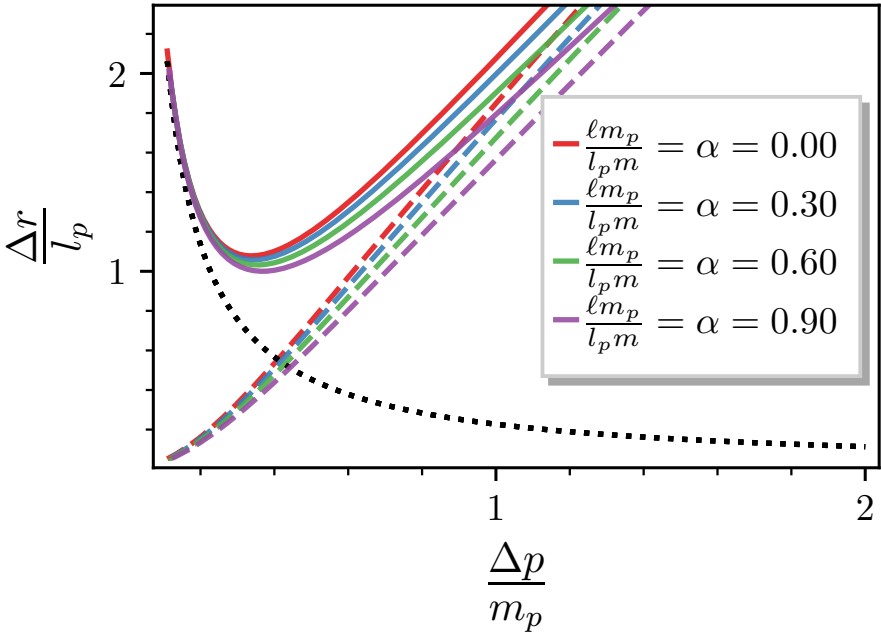

**Figure 5.** GUP profile emerged from the horizon wave function formalism for $\epsilon = 1$. The dotted line represents the particle size uncertainty $\Delta r_0$, the dashed line represents the uncertainty of the horizon radius $\Delta r_H$, and the solid lines describe the GUP.

## 4. Discussion

A few years ago, effective theories suggested lowering the scale of quantum black hole formation to TeV. Thus, in principle, it became experimentally accessible. In spite of no quantum black holes being detected, solid theoretical results point out that such objects should exist in nature [7,14]. They could give us valuable hints about quantum gravity features [7,13,14]. One of this paper's motivating questions was whether a generic black hole hair could significantly change the scale of quantum black hole formation. However, regarding the analysis carried out here, the hairy black holes look qualitatively similar to the Schwarzschild one, with a probability $P_{BH}$ of a similar shape and a related GUP, leading to the existence of a minimum length scale. Nevertheless, one of the main results of the present paper is that the existence of hair increases the probability $P_{BH}$. This is indeed a point to be stressed. Its explanation rests upon the fact that the hairy black hole radius is slightly larger than the one for Schwarzschild. This implies that, although the scale of quantum black hole formation is still beyond the current experimental scale, additional fields may lower such scale. Those results might impact future colliders' estimations of quantum black holes coming from alternative theories of gravity and potentially stimulate investigations of specific models of quantum hairy black holes [17].

**Author Contributions:** Conceptualization, R.T.C.; Formal analysis, R.T.C. and J.M.H.d.S.; Investigation, R.T.C. and J.M.H.d.S.; Writing—original draft, R.T.C. and J.M.H.d.S. All authors have read and agreed to the published version of the manuscript.

**Funding:** This research was funded by Unesp | AGRUP grant PROPe 13/2022 and CNPq grant No. 303561/2018-1.

**Institutional Review Board Statement:** Not applicable.

**Informed Consent Statement:** Not applicable.

**Data Availability Statement:** Not applicable.

**Acknowledgments:** R.T.C. thanks Unesp | AGRUP for the financial support. J.M.H.d.S. thanks CNPq (grant No. 303561/2018-1) for the financial support.

**Conflicts of Interest:** The authors declare no conflict of interest.

## Note

[1] The analytical expression of $\Delta r_{\mathrm{H}}^2$ is huge and little enlightening.

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
