# Peer review of "Quantum Hairy Black Hole Formation and Horizon Quantum Mechanics"

_universe, doi:10.3390/universe9010023_

Round 1

Reviewer 1 Report

In this work the authors address the problem of the formation of a hairy black hole obtained through gravitational decoupling by applying the horizon quantum mechanics formalism. The results are interesting to some extent and I would recommend the paper for publication after the authors take into account the following points.

1. Page 2 line 55 reads “conservation equation must be applied”. This phrase is confusing. Indeed, as the Einstein tensor es covariantly conserved, the energy momentum tensor sourcing the geometry must be conserved.

2. The authors are encouraged to discuss the consequences of the decoupling in the conservation of the energy momentum tensor for each source. As it is well established, when the geometric deformation is minimal (only the radial component is deformed) both sources are independently conserved. However, when the deformation involved both metric function, the sources exchange energy-momentum. There are some examples of how is such exchange of energy in the literature and the authors must cite such a works.

3. In section 3 the authors must include some references as a guide of the discussion about the horizon wave function formalism.

4. The author might discuss to what extent their results are highly dependent on the particular form of the wave function.

Author Response

We thank the referee for the valuable comments/suggestions. As highlighted in the text, all the points were addressed, as well as the points raised by the other refereees, within the new manuscript version. We also reviewed the whole text, correcting misprints and improving its linguistics.

Reviewer 2 Report

The theme of the manuscript is of current demand and the authors have written it nicely. However, before recommending it for publication I would like to suggest the authors to take care of the following minor corrections:

(1) In Eqs. (14) and (16) no where the meaning of "e" has been expressed explicitly.

(2) There are several grammatical errors in the entire manuscript, e.g. in line nos. 43 & 44 , 76, in Figures 3 and 4 , line no. 109.

(3) In figures, how the numerical values for "l" and "\alpha" have been chosen-notes in the relevant places are needed.

(4) Please add some more in the last section 4 (Discussion) to explain the clear picture of the findings in the paper.

Author Response

(The authors gave the same response as above.)

Reviewer 3 Report

In this paper, the author presented the formation of quantum hairy black holes by applying the
horizon quantum mechanics formalism and the gravitational decoupling method. A graphical presentation has been given for the probability of quantum black hole formation and hairy black holes for a range of hair parameters. The calculations seem to be straightforward and it seems too many assumptions as a realistic cosmological model.

Regarding the novelty and significance of the results, I consider the contribution to be marginally relevant. In comparison with the past studies, the new points and important astrophysical consequences results should be explained more explicitly. That is, the originality of this work must be clarified.

Finally, I suggest that if possible, try to combine the results with current astronomical observations.

I request the author to double-check the typographical and grammatical mistakes.

Author Response

We thank the referee for the valuable comments/suggestions. We also want to clarify that our results involved no cosmological assumptions. Furthermore, the results are not related to astrophysical black holes. Those points were highlighted in sections I and IV, as well as the points raised by the other refereees. We also reviewed the whole text, correcting misprints/grammatical mistakes and improving the discussion about the results.

Reviewer 4 Report

This paper presents a model for quantum hairy black hole formation in the context of quantum mechanics. The paper is mostly well written and presents sounds results. I include some suggestions for improvements.

- Given that this is quite a specialised field it would be good it the authors could provide some more context to interpret their results. Is this model aim at understanding microscopic (Planck scale)  black hole formation through quantum fluctuations?  If so, what are the expected physical/experimental implications? Why is this relevant?

- Explain the kind of hair associated to the model. How does Theta in Eq.3 relate to hair parameters in Eq.15? Is this also a quantum field that fluctuates?

- How is Eq.22 an energy spectrum? or is it just the minimum energy E0?

- How the wave function in Eq.24 is a function of r_H? and of nothing else? Is horizon wave function the same as minimum energy wave function? 

- Why the wave function in Eq.29 is not a function of time? In what sense is a wave function?

- Provide a reference or further explanation for the last sentence in the Discussion. The paper talks about Planck fluctuations, how si that relevant to current or future colliders, which operate at much lower energies?

Author Response

We thank the referee for the valuable comments/suggestions. The points raised are discussed below. Also, as highlighted in the text, all the points raised by the other referees are addressed within the new manuscript version.

  • Yes, the aim of HQM is understanding microscopic black hole formation. The stating point is to introduce uncertainty in the horizon radius, as in any quantum observable. It could be very relevant for testing the reasonability of microscopic black hole formation in a given scale, not necessarily the Plank scale, for different effective theories. A comprehensive review of the formalism is found in reference [46], as highlighted at the beginning of Section III of the present version of the manuscript.

  • The source Theta is the origin, along with the geometric deformation and the strong energy condition, of the hair parameters. The details are found in reference [26] of the present manuscript. No model is assumed for Theta, nor is its origin fixed a priori, only that it has the same symmetry as the deformed metric. Thus, in principle, it could be related to a quantum field that fluctuates.

  • Eq.22 is just the expression of the Schwarzschild radius in terms of Plank constants and taking E equals the black hole mass. It was introduced here for ilustrating the basic setup of the formalism. In our results we have used Eq. 16 instead.

  • The wave function in Eq.24 is turned into a function of r_H by using the expression of the black hole radius, with E function of r_H as explained in the above point. The horizon wave function is the wave function associated to the horizon radius, which depends on the energy of the quantum state.

  • In the present version, the formalism dos not give the dynamics of the object described by the wave function. The probability of formation comes from combining the hoop conjecture with the probability of the object being encompassed by an horizon of radius r_H, as described in the above points.

  • The section Discussion was improved and the mentioned sentence rewritten.

We also reviewed the whole text, correcting misprints/grammatical mistakes and improving the discussion about the results.